# INSTANCE SEGMENTATION WITH SUPERVOXEL BASED TOPOLOGICAL LOSS FUNCTION

## ABSTRACT

Reconstructing the intricate local morphology of neurons as well as their long-range projecting axons can address many connectivity related questions in neuroscience. While whole-brain imaging at single neuron resolution has recently become available with advances in light microscopy, segmenting multiple entangled neuronal arbors remains a challenging instance segmentation problem. Split and merge mistakes in automated tracings of neuronal branches can produce qualitatively different results and represent a bottleneck of reconstruction pipelines. Here, by extending the notion of simple points from digital topology to connected sets of voxels (i.e. supervoxels), we develop a topology-aware neural network based segmentation method with minimal overhead. We demonstrate the merit of our approach on a newly established public dataset that contains 3-d images of the mouse brain where multiple fluorescing neurons are visible as well as the DRIVE 2-d retinal fundus images benchmark.

## 1 INTRODUCTION

High-throughput reconstruction of neurons is a challenging 3-d instance segmentation problem and represents the bottleneck of many data-driven neuroscience studies (Winnubst et al., 2019; Gouwens et al., 2020). In recent years, deep learning-based methods have become the leading framework for segmenting individual neurons which is the first step towards reconstructing neural circuits (Turaga et al., 2010; Januszewski et al., 2018; Lee et al., 2019). While these studies and others have significantly improved the quality, automated segmentations of neurons from large 3-d volumes still contain many topological mistakes (i.e. splits and merges) and require extensive human proofreading. This can be attributed to two basic observations: (i) neuronal branches can be as thin as the size of a single voxel, (ii) branches often appear to touch or even overlap due to the imaging resolution and artifacts. Consequently, seemingly innocuous mistakes at the single voxel level can produce catastrophically incorrect segmentations.

A natural solution to these problems is to take the topology of the underlying objects into account during the training process. In digital topology, a simple voxel of a binary 3-d image is defined as a foreground voxel whose deletion does not change the topology of the image (Kong & Rosenfeld, 1989). (i.e., does not cause splits/merges, create/delete loops, holes, objects) Accuracy in segmenting simple voxels is, therefore, inconsequential from the perspective of topological correctness. An efficient method for identifying such voxels (Bertrand & Malandain, 1994) was utilized in warping the reference segmentation to flip noisy labels at object boundaries in electron microscopy images (Jain et al., 2010). This characterization was also used to place more emphasis on non-simple voxels to segment neurons in light microscopy images (Gornet et al., 2019).

However, multiple connected voxels are involved in most topological mistakes, which is ignored in the voxel-based perspective. Therefore, we first pursue extending the theory of simple voxel characterization to supervoxels (i.e., connected components). We then use this theory to propose efficient methods to characterize the topological role of supervoxels in biomedical instance segmentation problems. We propose a simple, differentiable cost function based on this supervoxel characterization to enable training of neural networks to minimize split/merge mistakes efficiently. Finally, we test our approach on 3-d images of the mouse brain that label multiple neurons, obtained by lightsheet microscopy as well as 2-d fundus images of the human retina Staal et al. (2004), to

demonstrate its merit in decreasing topological mistakes with minimal overhead during training of the neural network.

## 2 RELATED WORKS

Accurate segmentation of fine-scale structures, e.g., neurons, vessel, and roads from satellite images is a challenging problem that has been intensely studied. There are numerous methods that aim to accurately reconstruct an object's topology by either incorporating topology-inspired loss functions during training or learning better feature representations (see Hu et al. (2023); Mosinska et al. (2018); Reininghaus et al. (2014); Sheridan et al. (2023); Wu et al. (2017). In addition, there have also been several works that utilize homotopy warping to emphasize mistakes at non-simple pixels as opposed to noncritical boundary differences (Hu, 2022; Jain et al., 2010).

Topology-inspired loss functions identify topologically critical locations where the neural network is error-prone, then enforce improvement via gradient updates. Turaga et al. (2009) developed MALIS which aims to improve the network's output at maximin edges. Each gradient update involves an expensive maximin search in a restricted window to find the voxels that are most prone to introducing topological mistakes in order to learn from those examples (Funke et al., 2018; Turaga et al., 2009). Gornet et al. (2019) leveraged digital topology to place higher penalties on incorrectly predictions at non-simple voxels. Shit et al. (2021) utilized morphological skeletons to compute a connectivity-aware loss function based on the Dice coefficient.

Clough et al. (2019) utilized Betti numbers of the ground truth as a topological prior, then computed gradients that increase or decrease the persistence of topological features in the prediction. Hu et al. (2019) penalized differences between the persistence diagrams of the prediction and ground truth. This involves an expensive search to find an optimal correspondence between persist features. In a more recent work, Hu et al. (2021) used discrete Morse theory to detect topologically critical structures, then used a persistence-based pruning scheme to filter them.

## 3 METHOD

Let $G = (V, E)$ be an undirected graph with the vertex set $V = \{1, \ldots, n\}$. We assume that $G$ is a graphical representation of an image where the vertices represent voxels and edges are defined with respect to a $k$-connectivity[1] constraint. A ground truth segmentation $y = (y_1, \ldots, y_n)$ is a labeling of the vertices such that $y_i \in \{0, 1, \ldots, m\}$ denotes the label of node $i \in V$. Each segment has a label in $\{1, \ldots, m\}$ and the background is marked with 0.

Let $F(y)$ be the foreground of the vertex labeling such that $F(y) = \{i \in V : y_i \neq 0\}$. Note that the foreground may include multiple, potentially touching objects. Let $\mathcal{S}(y) \subseteq \mathcal{P}(V)$ be the set of connected components induced by the labeling $y$, where $\mathcal{P}(V)$ denotes the power set of $V$. Let $\hat{y} = (\hat{y}_1, \ldots, \hat{y}_n)$ be a prediction of the ground truth such that $\hat{y}_i \in \{0, 1, \ldots, \ell\}$, where $\ell$ is the number of objects in the predicted segmentation[2].

In a labeled graph, the connected components are determined by the equivalence relation that $i \sim j$ if and only if $y_i = y_j$ with $i, j \in F(y)$ and there exists a path from $i$ to $j$ that is entirely contained within the same segment. An equivalence relation induces a partition over a set into equivalence classes which correspond to the connected components in this setting.

We propose a novel topological loss function to train a neural network with the goal of avoiding false merges between, and false splits of, the foreground objects.

**Definition 1.** *Let* $\mathcal{L} : \mathbb{R}^n \times \mathbb{R}^n \to \mathbb{R}$ *be the topological loss function given by*

$$\mathcal{L}(y, \hat{y}) = \mathcal{L}_0(y, \hat{y}) + \alpha \sum_{C \in \mathcal{N}(\hat{y})} \mathcal{L}_0(y_C, \hat{y}_C) + \beta \sum_{C \in \mathcal{P}(\hat{y})} \mathcal{L}_0(y_C, \hat{y}_C)$$

*such that* $\alpha, \beta \in \mathbb{R}_+$ *and* $\mathcal{L}_0$ *is an arbitrary loss function.*

---

[1] We assume that $k \in \{4, 8\}$ and $k \in \{6, 18, 26\}$ for 2D and 3D images, respectively. See Kong & Rosenfeld (1989) for basic definitions of connectivity between voxels in an image.

[2] We assume that the true number of objects $m$ is unknown at the time of inference.

We build upon a traditional loss function $\mathcal{L}_0$ (e.g. cross-entropy or Dice coefficient) by adding additional terms that penalize sets of connected voxels (i.e. supervoxels) that cause topological mistakes. These supervoxels are formed by computing connected components of the false negative and false positive masks, respectively (i.e., by considering foregrounds of $y, \hat{y}$). The sets $\mathcal{N}(\hat{y}_-)$ and $\mathcal{P}(\hat{y}_+)$ contain components whose removal or addition, respectively, changes the number of connected components. A component that changes the underlying topology in this manner is referred to as a *critical* component. The objective this section is to rigorously define these sets, then present an algorithm that detects such components.

## 3.1 CRITICAL COMPONENTS

Critical components are an extension of the notion of non-simple voxels from digital topology to connected sets (i.e. supervoxels). Intuitively, a voxel is called non-simple if its removal or addition changes the number of connected components, holes, or cavities. Analogously, a supervoxel is called critical if its removal or addition changes the topology. We use the terminology *critical* as opposed to *non-simple* since the definition is not a direct generalization for computational reasons. The advantage of limiting the definition is that it enables our supervoxel-based loss function to be computed in linear time, which is a significant improvement over related topological loss functions.

### 3.1.1 FALSE SPLITS

Let $\hat{y}_-$ be the false negative mask determined by comparing the prediction to the ground truth. Let $\mathcal{S}_y(\hat{y}_-)$ be the set of connected components of $\hat{y}_-$ with respect to $y^3$. In this definition, the connected components are determined by the equivalence relation that $i \sim j$ if and only if $(\hat{y}_-)_i = (\hat{y}_-)_j$ and $y_i = y_j$ with $i, j \in F(\hat{y}_-)$ in addition to the existence of a path from $i$ to $j$ contained within the same segments. The second condition guarantees that each component in the false negative mask corresponds to only one component in the ground truth.

*Negatively* critical components are determined by comparing the number of connected components in $y$ and $y \ominus C$. The notation $y \ominus C$ denotes "removing" a component from the ground truth. The result of this operation is a vertex labeling where the label of node $i \in V$ is given by

$$(y \ominus C)_i = \begin{cases} 0, & \text{if } i \in C \\ y_i, & \text{otherwise} \end{cases} \tag{1}$$

The removal of a component only impacts a specific region within the graph; the component itself and the nodes connected to its boundary. Thus, for topological characterization, it is sufficient to check whether the removal changes the number of connected components in that region (Bertrand & Malandain, 1994). Let $N(C) \subseteq V$ be the neighborhood surrounding a component $C \in \mathcal{S}(y)$ such that $N(C) = \{i \in V : \{i, j\} \in E \text{ and } j \in C\}$. Let $y \cap N(C)$ represent the labeling $y$ within $N(C)$.

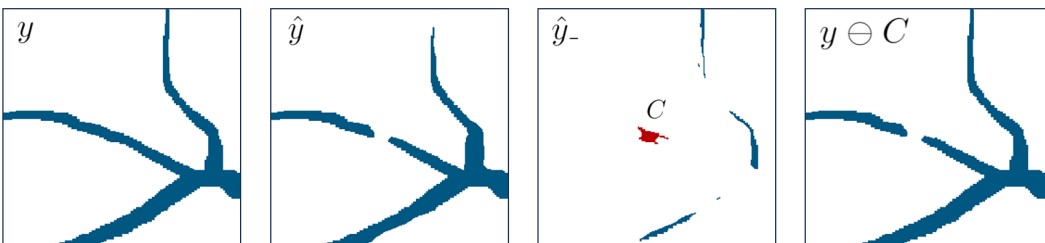

Figure 1: **Left:** patches of ground truth and predicted segmentations. **Third:** false negative mask with component $C$ highlighted. **Right:** $C$ is negatively critical since its removal changes topology.

**Definition 2.** *A component $C \in \mathcal{S}_y(\hat{y}_-)$ is said to be negatively critical if $|\mathcal{S}(y \cap N(C))| \neq |\mathcal{S}((y \ominus C) \cap N(C))|$.*

Negatively critical components change the local topology by either deleting an entire component or altering the connectivity between vertices within $N(C)$. In the latter case, the removal of such

---

[3]Note that $\mathcal{S}_y(y) = \mathcal{S}(y)$ in the special case when the argument and subscript are identical.

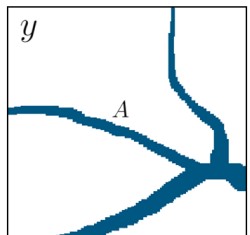 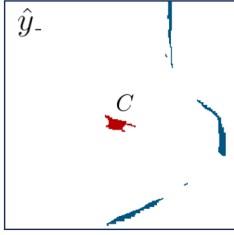 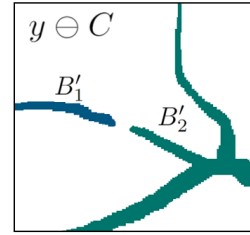 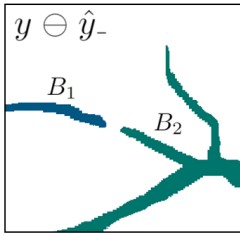

Figure 2: Visualization of Corollary 1.

components locally disconnects some $i, j \in N(C)$ so that it is impossible to find a path (in this neighborhood) that does not pass through $C$. Based on this intuition, we can establish an equivalent definition of negatively critical components as components that (1) are identical to a component in the ground truth, or (2) locally disconnect at least one pair of nodes in $N(C)$ after being removed.

**Theorem 1.** *A component $C \in \mathcal{S}_y(\hat{y}_-)$ is negatively critical if and only if there exists an $A \in \mathcal{S}(y \cap N(C))$ with $A \supseteq C$ such that either: (1) $A = C$ or (2) $\exists v_0, v_k \in A \setminus C$ such that there does not exist a path $(v_0, \ldots, v_k) \subseteq N(C)$ with $v_i \notin C$ for $i = 1, \ldots, k-1$. (Proof is in Append A.1)*

A computational challenge in both definitions is the need to recompute connected components within the neighborhood $N(C)$ for every $C \in \mathcal{S}_y(\hat{y}_-)$. In the worst case, the computational complexity is $\mathcal{O}(n^2)$ with respect to the number of voxels in the image. However, we can develop a more efficient algorithm with $\mathcal{O}(n)$ complexity by leveraging two useful facts: (1) neurons are tree-structured objects, implying that, (2) negatively critical components change both the local *and* global topology.

Recall that a negatively critical component $C \in \mathcal{S}_y(\hat{y}_-)$ changes the local topology of $N(C)$ in the sense that $|\mathcal{S}(y \cap N(C))| \neq |\mathcal{S}((y \ominus C) \cap N(C))|$. Analogously, $C$ also changes the global topology if $|\mathcal{S}(y)| \neq |\mathcal{S}(y \ominus C)|$. In this special case, we can establish an equivalent definition, similar to Theorem 1, that utilizes $\mathcal{S}(y)$ and $\mathcal{S}(y \ominus C)$ in place of $\mathcal{S}(y \cap N(C))$ and $\mathcal{S}((y \ominus C) \cap N(C))$.

However, this characterization can be streamlined by incorporating $\mathcal{S}(y \ominus \hat{y}_-)$ instead of $\mathcal{S}(y \ominus C)$, where $y \ominus \hat{y}_-$ denotes the ground truth after removing every component in the false negative mask:

$$(y \ominus \hat{y}_-)_i = \begin{cases} 0, & \text{if } (\hat{y}_-)_i = 1 \\ y_i, & \text{otherwise} \end{cases}$$

**Corollary 1.** *A component $C \in \mathcal{S}_y(\hat{y}_-)$ is negatively critical with $|\mathcal{S}(y)| \neq |\mathcal{S}(y \ominus C)|$ if and only if there exists an $A \in \mathcal{S}(y)$ with $A \supseteq C$ such that either: (1) $A = C$ or (2) $\exists B_1, B_2 \in \mathcal{S}(y \ominus \hat{y}_-)$ with $B_1, B_2 \subset A$ such that $B_1 \cup C \cup B_2$ is connected. (Proof is in Append A.2)*

### 3.1.2 FALSE MERGES

Let $\hat{y}_+$ be the false positive mask determined by comparing the prediction to the ground truth. A component in the false positive mask is positively critical if its addition to the ground truth changes the topology. Equivalently, removing such component from the prediction changes the topology in this image. For computational reasons, we use this definition because it streamlines adapting results from the previous section.

**Definition 3.** *A component $C \in \mathcal{S}_y(\hat{y}_+)$ is said to be positively critical if $|\mathcal{S}(\hat{y} \cap N(C))| \neq |\mathcal{S}(\hat{y} \ominus C \cap N(C))|$.*

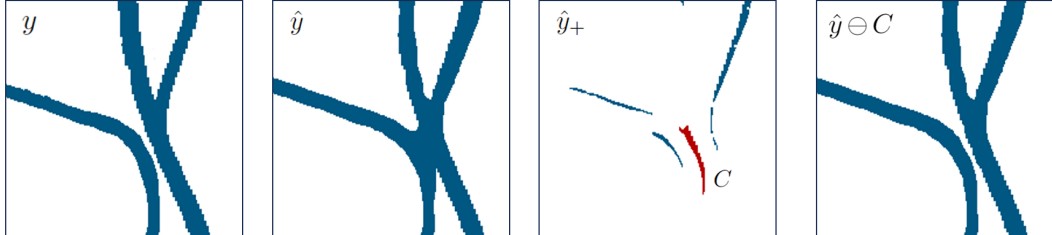

Figure 3: **Left:** patches of ground truth and predicted segmentations. **Third image:** false positive mask with a single component $C$ highlighted. **Right:** $C$ is positively critical since its removal changes the number of connected components in the prediction.

Positively critical components change the local topology by either (1) creating a component or (2) altering the connectivity between distinct locally connected components in the ground truth. For the latter case, these components connect certain pairs of nodes that belong to locally distinct components. Equivalently, the removal of these components from the prediction causes certain pairs of nodes to become locally disconnected. Next, we present an equivalent definition that characterizes positively critical components as satisfying one of these conditions.

**Theorem 2.** *A component $C \in \mathcal{S}_y(\hat{y}_+)$ is positively critical if and only if there exists an $A \in \mathcal{S}(\hat{y})$ with $A \supseteq C$ such that either:* (1) $A = C$ *or* (2) $\exists\, v_0, v_k \in A \setminus C$ *such that there does not exist a path* $(v_0, \dots, v_k) \subseteq N(C)$ *with* $v_i \notin C$ *for* $i = 1, \dots, k-1$. *(Proof is in Appendix A.1)*

Similarly, positively critical components present the same computational challenge of needing to recompute connected components for every $C \in \mathcal{S}_y(\hat{y}_+)$. However, we can avoid this expensive calculation by utilizing a corollary of Theorem 2 that establishes an equivalent definition of positively critical components that also change the global topology. This characterization uses $\mathcal{S}(\hat{y})$ and $\mathcal{S}(\hat{y} \ominus \hat{y}_+)$ (instead of $\mathcal{S}(y \cap N(C))$ and $\mathcal{S}(\hat{y} \ominus C \cap N(C))$), where $\hat{y} \ominus \hat{y}_+$ denotes removing every component in the false positive mask from the prediction via

$$(\hat{y} \ominus \hat{y}_+)_i = \begin{cases} 0, & \text{if } (\hat{y}_+)_i = 1 \\ \hat{y}_i, & \text{otherwise} \end{cases}$$

**Corollary 2.** *A component $C \in \mathcal{S}_y(\hat{y}_+)$ is positively critical with $|\mathcal{S}(\hat{y})| \neq |\mathcal{S}(\hat{y} \ominus C)|$ if and only if there exists an $A \in \mathcal{S}(\hat{y})$ with $A \supseteq C$ such that either:* (1) $A = C$ *or* (2) $\exists B_1, B_2 \in \mathcal{S}(\hat{y} \ominus \hat{y}_+)$ *with $B_1, B_2 \subset A$ such that $B_1 \cup C \cup B_2$ is connected. (Proof is in Appendix A.2)*

### 3.2 COMPUTING CRITICAL COMPONENTS

Although topological loss functions improve the segmentation quality, a major drawback is that they are computationally expensive. A key advantage of our proposed method is that the runtime is $\mathcal{O}(n)$ with respect to the number of voxels. In contrast, the runtime of related methods is typically either $\mathcal{O}(n \log n)$ or $\mathcal{O}(n^2)$ (e.g. Jain et al. (2010); Turaga et al. (2009); Gornet et al. (2019); Hu et al. (2021); Shit et al. (2021); Hu et al. (2023)).

In the case of identifying non-simple voxels, Bertrand & Malandain (1994) prove that it is sufficient to examine the topology of the neighborhood. Similarly, we can determine whether a component is critical by checking the topology of nodes connected to the boundary. For the remainder of this section, we focus the discussion on computing negatively critical components since the same algorithm can be used to compute positively critical components.

Let $D(C) = N(C) \setminus C$ be the set of nodes connected to the boundary of a component $C \in \mathcal{S}_y(\hat{y}_-)$. Assuming that a negatively critical component also changes the global topology, Corollary 1 can be used to establish analogous conditions on the set $D(C)$ that are useful for computation.

**Corollary 3.** *A component $C \in \mathcal{S}_y(\hat{y}_-)$ is negatively critical with $|\mathcal{S}(y)| \neq |\mathcal{S}(y \ominus C)|$ if and only if $\exists A \in \mathcal{S}(y)$ with $A \supseteq C$ such that either:* (1) $\nexists i \in D(C)$ *with $i \in A$ or* (2) $\exists B_1, B_2 \in \mathcal{S}(y \ominus \hat{y}_-)$ *with $B_1, B_2 \subset A$ such that $i \in B_1$ and $j \in B_2$ for some $i, j \in D(C)$. (Proof is in Appendix A.3)*

Using Corollary 3 to compute critical components involves: (i) computing sets of connected components and (ii) checking Conditions 1 and 2. The key to performing this computation in linear time

is to first precompute $\mathcal{S}(y)$ and $\mathcal{S}(y \ominus \hat{y}_-)$, then compute $\mathcal{S}_y(\hat{y}_-)$ with a breadth-first search (BFS) while simultaneously checking whether Conditions 1 or 2 hold. Intuitively, the crux of this last step is to leverage that a BFS terminates once the search reaches the boundary of a connected component. Since the set $D(C)$ is connected to the boundary, each node in this set will be visited.

Let $r \in F(\hat{y}_-)$ be the root of the BFS. Given a node $j \in D(C)$, Conditions 1 and 2 can be checked with a hash table called *collisions* that stores the connected component label of $j$ in $\mathcal{S}(y)$ and $\mathcal{S}(y \ominus \hat{y}_-)$ as a key-value pair, respectively. If we never visit a node $j \in D(C)$ with the same ground truth label as the root, then this label is not a key in collisions and so the component satisfies Condition 1 (see Line 19 in Algorithm 2).

Now consider the case when we do visit a node $j \in D(C)$ with the same ground truth label as the root. A new entry is created in the hash table if this label is not a key (see Line 15 in Algorithm 2). Otherwise, the value corresponding to this key is compared to the label of $j$ in $\mathcal{S}(y \ominus \hat{y}_-)$. If these labels differ, then the connected component satisfies Condition 2. We provide pseudo code for this method in Algorithms 1 and 2.

**Theorem 3.** *The computational complexity of computing critical components that satisfy either $|\mathcal{S}(y)| \neq |\mathcal{S}(y \ominus C)|$ or $|\mathcal{S}(\hat{y})| \neq |\mathcal{S}(\hat{y} \ominus C)|$ is $\mathcal{O}(n)$ with respect to the number of voxels in the image. (Proof is in Appendix A.3)*

We emphasize the the statements and algorithms surrounding Theorem 3 are restricted to tree-structured objects (i.e. critical components that satisfy $|\mathcal{S}(y)| \neq |\mathcal{S}(y \ominus C)|$ or $|\mathcal{S}(\hat{y})| \neq |\mathcal{S}(\hat{y} \ominus C)|$). Indeed, a similar algorithm based on the main definitions and deductions can be implemented in a straightforward way, except that this algorithm will be super-linear in complexity.

---

**Algorithm 1** Detection of Critical Components

```
 1: procedure DETECT_CRITICALS(y, ŷ):
 2:     ŷ₋ ← compute false negatives
 3:     S(y) ← compute connected components
 4:     S(y ⊖ ŷ₋) ← compute connected components
 5:     N(ŷ) = get_critical(y, ŷ₋, S(y), S(y ⊖ ŷ₋))
 6:
 7:     ŷ₊ ← compute false positives
 8:     S(ŷ) ← compute connected components
 9:     S(ŷ ⊖ ŷ₊) ← compute connected components
10:     P(ŷ) = get_critical(ŷ, ŷ₊, S(ŷ), S(ŷ ⊖ ŷ₊))
11:     return N(ŷ), P(ŷ)
12: end procedure
13:
14: # Note that ŷ× is a placeholder for ŷ₋ and ŷ₊
15: procedure GET_CRITICAL(y, ŷ×, S(y), S(y ⊖ ŷ×))
16:     F(ŷ×) ← compute foreground
17:     X(ŷ×) = set()
18:     while |F(ŷ×)| > 0 :
19:         r = sample(F(ŷ×))
20:         C, is_critical = get_component(y, ŷ×, S(y), S(y ⊖ ŷ×), r)
21:         F(ŷ×).remove(C)
22:         if is_critical :
23:             X(ŷ×).add(C)
24:     return X(ŷ×)
25: end procedure
```

---

### 3.3 PENALIZING CRITICAL TOPOLOGICAL MISTAKES

Our topological loss function builds upon classical, voxel-based loss functions by adding terms that penalize critical components. The paradigm shift here is to evaluate each mistake at a "structure level" that transcends rectilinear geometry as opposed to the voxel level. In standard loss functions, mistakes are detected at the voxel-level by directly comparing the prediction at each voxel against

---

**Algorithm 2** Check if Component is Critical

---

1: **procedure** GET_COMPONENT($y$, $\hat{y}_\times$, $\mathcal{S}(y)$, $\mathcal{S}(y \ominus \hat{y}_\times)$, $r$ ):
2:     $C = \text{set}()$
3:     $collisions = \text{dict}()$
4:     $is\_critical = \text{False}$
5:     $queue = [r]$
6:     **while** $|queue| > 0$ :
7:         $i = queue.\text{pop}()$
8:         $C.\text{add}(i)$
9:         **for** $j$ in $N(i)$:
10:            **if** $y_j == y_r$:
11:                **if** $(\hat{y}_\times)_j == 1$:
12:                    $queue.\text{push}(j)$
13:                **else**:
14:                    $\ell_j = \text{get\_label}(\mathcal{S}(y), j)$
15:                    **if** $\ell_j$ not in $collisions.\text{keys}()$:
16:                        $collisions[\ell_j] = \text{get\_label}(\mathcal{S}(y \ominus \hat{y}_\times), j)$
17:                    **elif** $collisions[\ell_j] \,!= \text{get\_label}(\mathcal{S}(y \ominus \hat{y}_\times), j)$:
18:                        $is\_critical = \text{True}$
19:        **if** $y_r$ not in $collisions.\text{keys}()$ :
20:            $is\_critical = \text{True}$
21:        **return** $C$, $is\_critical$
22: **end procedure**

---

the ground truth. Instead, we consider the context of each mistake by determining whether a given supervoxel causes a critical topological mistake.

One advantage of our topological loss function is that it is architecture agnostic and can be easily integrated into existing neural network architectures. We first train a baseline model with a standard loss function, then fine-tune with the topological loss function once the performance of the baseline model starts to plateau. This could be achieved gradually by integrating the topological loss function into the model with a continuation scheme over $n$ epochs,

$$\mathcal{L}(y, \hat{y}, i) = (1 - t_i)\,\mathcal{L}_0(y, \hat{y}) + \alpha\,t_i\,\mathcal{L}_-(y, \hat{y}) + \beta\,t_i\,\mathcal{L}_+(y, \hat{y})$$

where $t_i = \min(i/n, 1)$ and $i$ is the current epoch.

The objective of hyperparameter optimization is to minimize the number of critical mistakes. The hyperparameters $\alpha, \beta \in \mathbb{R}$ are scaling factors that control how much weight is placed on splits versus merges. When it is preferable to avoid false merges (e.g., when they are more difficult to detect and time consuming to fix), one can prioritize learning to avoid this type of mistake by setting $\beta > \alpha$ so that merges receive higher penalties.

Our topological loss function adds little computational overhead since the only additional calculation is computing the critical components. In Theorem 3, we prove that Algorithms 1 and 2 can be used to compute critical components in linear time. This result can then be used to show that the computational complexity of computing $\mathcal{L}$ is also $\mathcal{O}(n)$.

## 4 EXPERIMENTS

We evaluate our method on two biomedical image datasets: **EXASPIM**[4] and **DRIVE**[5]. The first consists of 3-d images of multiple neurons that were generated with Expansion-Assisted light Selective Plane Illumination Microscope (ExA-SPIM) (Glaser et al., 2023). This dataset consists of 37 volumetric images whose sizes range from 256x256x256 to 1024x1024x1024 and voxel size is $\sim 1\,\mu\text{m}^3$. DRIVE is a retinal vessel segmentation dataset consisting of 20 images with a size of 584x565 (Staal et al., 2004).

---

[4]Downloaded from the AWS bucket s3://aind-msma-morphology-data/EXASPIM
[5]Downloaded from https://drive.grand-challenge.org

**Hyperparameter optimization.** An important component of our proposed method are the hyperparameters $\alpha$ and $\beta$ that scale how much weight is placed on critical mistakes. In general, we have observed that $\alpha, \beta > 0$ improves topological accuracy. In order to gain insight on how these parameters affect the performance of a model, we performed hyperparameter optimization (Akiba et al., 2019). In this experiment, we trained a U-Net on the EXASPIM dataset and used edge accuracy as the objective function. We found that the performance changes less than $10\%$ across changes of a few orders of magnitude around the optimal values of the hyperparameters as shown in Figure 4. Thus, these experiments suggest that careful hyperparameter tuning is not necessary in practice. In fact, the results obtained by our method in Table 2 were achieved without using hyperparameter optimization.

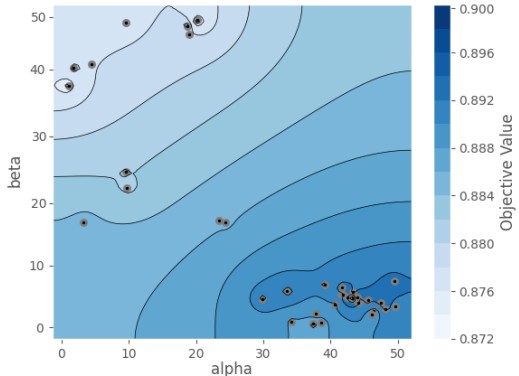

Figure 4: Objective function, each black dot is a trial.

**Performance metrics.** In neuron segmentation, the ultimate goal is to reconstruct the network topology of neural circuits. To this end, each prediction is skeletonized and so we use four skeleton-based metrics (Appendix C.2) that reflect to what extent the topology of a neuron is correctly reconstructed. For the retinal vessel segmentation dataset, we use four pixel-based evaluation metrics: pixel-wise accuracy, Dice coefficient, Adapted Rand Index (ARI), and Variation of Information (VOI).

**Baselines.** For neuron segmentation, we use a U-Net as the backbone and compare our method to two types of baselines: (1) Standard losses: **Cross Entropy** and **Dice Coefficient** and (2) topology aware losses: **clDice** (Shit et al., 2021), **Simple Voxel** (Gornet et al., 2019) and **MALIS** (Turaga et al., 2009). For the vessel segmentation task, we compare our method to **Dive** (Fakhry et al., 2016), **U-Net** (Ronneberger et al., 2015), **Mosin.** (Mosinska et al., 2018), **TopoLoss** (Hu et al., 2019), and **DMT** (Hu et al., 2023).

**Evaluation.** EXASPIM: there are 33 and 4 images in the train and test set. All methods were trained for 1500 epochs. In our method, the hyperparameters $\alpha$ and $\beta$ were tuned via Bayesian optimization (Akiba et al., 2019) prior to training. DRIVE: $\alpha = 5$ was chosen manually. ($\beta$ – merge mistakes – does not exist in the case of binary segmentation.) We used 3-fold cross validation and report the mean performance on the validation set. See Appendix for further details.

**Quantitative and qualitative results.** Table 1 shows the quantitative results for the different models on the EXASPIM dataset. Table 3 (Appendix) shows the results on each individual block from the test set. Note that bold numbers highlight the best value across all methods. Table 2 shows the quantitative results for the DRIVE dataset. Our proposed method significantly outperforms the other methods in terms of topological accuracy. Figure 5 shows qualitative results from models trained with our proposed loss function in which $\mathcal{L}_0$, $L_-$, and $\mathcal{L}_+$ are defined using cross entropy loss. Although the only difference between these two models is the addition of the topological terms $\mathcal{L}_-$ and $\mathcal{L}_+$, there is a clear difference in topological accuracy.

Table 1: Quantitative results for different models on the EXASPIM dataset

| Method | # Splits ↓ | # Merges ↓ | % Omit ↓ | % Merged ↓ | % Accuracy ↑ |
|---|---|---|---|---|---|
| Cross Entropy | 39.50±22.37 | 4.00±3.16 | 9.14±0.0830 | 13.18±9.63 | 74.68±16.40 |
| Dice | 38.25±37.12 | 4.25±5.68 | 16.83±14.95 | 14.10±14.49 | 69.08±26.77 |
| clDice | 42.25±57.04 | **2.5±2.06** | 6.49±6.08 | 6.13±3.61 | 89.91±7.82 |
| Simple Voxel | 40.00±39.04 | 5.00±5.15 | 8.88±5.10 | **1.83±1.97** | 89.30±7.06 |
| MALIS | 30.25±30.55 | 3.00±2.55 | 7.48±0.0520 | 5.08±0.0313 | 87.48±7.84 |
| **Ours** | **16.75±19.45** | 3.50±3.57 | **4.48±2.75** | 4.20±4.24 | **91.33±6.94** |

Table 2: Quantitative results for different models on the DRIVE dataset

| Method | Accuracy ↑ | Dice ↑ | ARI ↑ | VOI ↓ | Betti Error ↓ |
|---|---|---|---|---|---|
| DIVE | **0.9549±0.0023** | 0.7543±0.0008 | 0.8407±0.0257 | 1.936±0.127 | 3.276±0.642 |
| U-Net | 0.9452±0.0058 | 0.7491±0.0027 | 0.8343±0.0413 | 1.975±0.046 | 3.643±0.536 |
| Mosin. | 0.9543±0.0047 | 0.7218±0.0013 | 0.8870±0.0386 | 1.167±0.026 | 2.784±0.293 |
| TopoLoss | 0.9521±0.0042 | 0.7621±0.0036 | 0.9024±0.0113 | 1.083±0.006 | 1.076±0.265 |
| DMT | 0.9495±0.0036 | 0.7733±0.0039 | 0.9024±0.0021 | 0.876±0.038 | **0.873±0.402** |
| **Ours** | 0.9533±0.0015 | **0.8092±0.0118** | **0.9433±0.0017** | **0.479±0.014** | 0.944±0.269 |

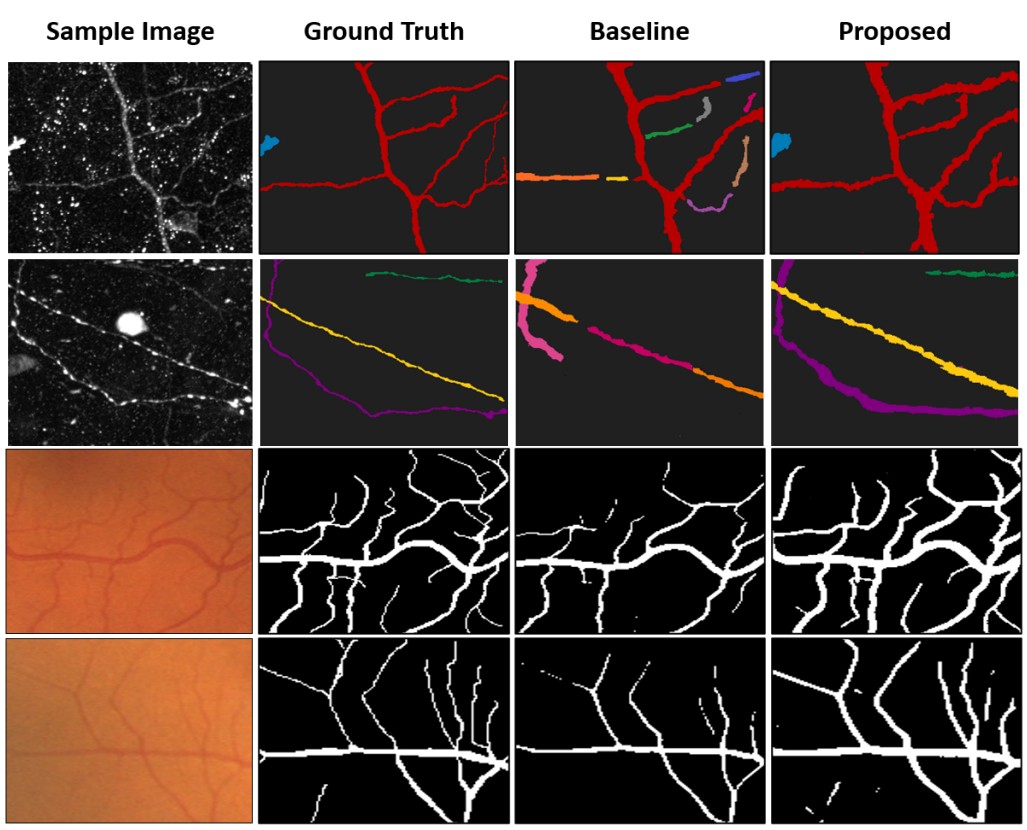

Figure 5: Qualitative results of the proposed method on the EXASPIM and DRIVE dataset. Baseline is a U-Net trained with cross entropy and Proposed is a U-Net trained with our supervoxel loss.

## 5 DISCUSSION

Mistakes that change the connectivity of the underlying objects (i.e., topological mistakes) are a key problem in instance segmentation. They produce qualitatively different results despite a few pixels/voxels being be incorrect, making it a challenge to avoid these mistakes with voxel-level objectives. Existing work on topology-aware segmentation typically requires costly steps to guide the segmentation towards topologically correct decisions, among other problems. Here, we developed a theoretical framework that generalizes the concept of simple voxel to connected components of arbitrary shape, and proposed a novel cost function with minimal computational overhead based on these results. We demonstrated our approach on two datasets with different resolution, dimensionality (2d vs 3d), and characteristics of the overall image content. Across multiple metrics, our method achieved state-of-the-art results.

It is now possible to image not only a local patch of neuronal morphology, but the whole arbor, which can reach multiple, distant brain regions. The favorable scalability of our approach will enable efficient analysis of such large datasets.

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
