## A    CHARACTERIZATIONS OF CRITICAL COMPONENTS

Given the graph $G = (V, E)$ and vertex labeling $y$, let $\mathcal{S}(y \cap N(C)) = \{A_1, \ldots, A_m\}$ be the connected components induced by this labeling with $A_i \subseteq F(y)$. Let $G' = (V', E')$ be the subgraph induced by this vertex labeling with $V' = \bigcup_{i=1}^{m} A_i$ and $E' = \bigcup_{i=1}^{m} E[A_i]$ where $E[A_i] = \{\{j, k\} \in E : j, k \in A_i\}$. Given a set $U \subseteq V'$, the subgraph induced by this set is $G'[U] = (U, E[U])$.

Let $\mu : \mathcal{P}(G') \to \mathbb{N}$ count the number of connected components in a given graph, where $\mathcal{P}(G')$ is the set of all subgraphs of a graph $G'$. We compute the disjoint union of subgraphs in the set $\mathcal{P}(G')$. Given the graphs $G_1 = (V_1, E_1)$ and $G_2 = (V_2, E_2)$, a disjoint union is defined as $G_1 \cup G_2 = (V_1 \cup V_2, E_1 \cup E_2)$.

**Lemma 1.** *$\mu$ has the following properties:*

*(i)* $\mu(G'[\varnothing]) = 0$.

*(ii)* *Non-negativity.* $\mu(G'[U]) \geq 0$ *for all* $U \subseteq V'$.

*(iii)* *Finite additivity. For any collection* $\{U_i\}_{i=1}^{n}$ *of pairwise disjoint sets with* $U_i \subseteq V'$,

$$\mu\left( \bigcup_{i=1}^{n} G'[U_i] \right) = \sum_{i=1}^{n} \mu(G[U_i]).$$

*Proof.* $\mu(G'[\varnothing]) = 0$ because the empty set does not contain any vertices. This function is non-negative by the definition of connected components. For finite additivity, a disjoint union over pairwise disjoint graphs does not affect the connectivity among vertices. Thus, this property holds by using basic set operations in an inductive argument. □

### A.1    GENERAL CASE

**Lemma 2.** *Given any component* $C \in \mathcal{S}_y(y_-)$*, there exists a unique* $A \in \mathcal{S}(y \cap N(C))$ *such that* $C \subseteq A$.

*Proof.* Choose any $j \in C$, then $y_j \neq 0$ by the definition of the false negative mask. Given that $j \in F(y)$, this implies that there exists some $A_i \in \mathcal{S}(y)$ with $j \in A_i$ and so $C \subseteq \cup_{i=1}^{m} A_i$. Using this inclusion, the set $C$ can be decomposed as

$$C = C \cap \bigcup_{i=1}^{m} A_i = \bigcup_{i=1}^{m} C \cap A_i.$$

This collection of sets is pairwise disjoint since $\{A_i\}_{i=1}^{m}$ is a collection of connected components.

Next, we claim that there exists a unique $A_j \in \mathcal{S}(y)$ such that $C \cap A_j \neq \varnothing$. By contradiction, suppose there exists a distinct $A_k \in \mathcal{S}(y)$ with $C \cap A_k \neq \varnothing$. But this assumption implies the existence of a path between $A_j$ and $A_k$ via $C$ because $C \subseteq F(y \cap N(C))$ along with $C \cap A_j \neq \varnothing$ and $C \cap A_k \neq \varnothing$. Since this contradicts $A_j$ and $A_k$ being disjoint, $A_j$ must be unique. Lastly, we can use this uniqueness property to conclude that

$$C = \bigcup_{i=1}^{m} C \cap A_i = C \cap A_j$$

which implies that $C \subseteq A_j$. □

**Theorem 1.** *A component* $C \in \mathcal{S}_y(\hat{y}_-)$ *is negatively critical if and only if there exists an* $A \in \mathcal{S}(y \cap N(C))$ *with* $A \supseteq C$ *such that either: (1)* $A = C$ *or (2)* $\exists\, v_0, v_k \in A \setminus C$ *such that there does not exist a path* $(v_0, \ldots, v_k) \subseteq N(C)$ *with* $v_i \notin C$ *for* $i = 1, \ldots, k-1$.

*Proof.* ($\Rightarrow$) Consider the case when $C \in \mathcal{S}_y(\hat{y}_-)$ is negatively critical due to $|\mathcal{S}(y \cap N(C))| > |\mathcal{S}((y \ominus C) \cap N(C))|$. Suppose that $\mathcal{S}(y \cap N(C)) = \{A_1, \ldots, A_m\}$, then starting from Equation 1

leads to the identity

$$
\begin{aligned}
|\mathcal{S}((y \ominus C) \cap N(C))| &= \mu\Big(\bigcup_{i=1}^{m} G'[A_i \setminus C]\Big) \\
&= \sum_{i=1}^{m} \mu(G'[A_i \setminus C]) \\
&= \sum_{\substack{i=1 \\ i \neq j}}^{m} \mu(G'[A_i]) + \mu(G'[A_j \setminus C]) \\
&= |\mathcal{S}(y \cap N(C))| - 1 + \mu(G'[A_j \setminus C]) \quad (2)
\end{aligned}
$$

where the second equality holds by $\mu$ being a finitely additive function defined over a collection of pairwise disjoint sets by Lemma 1. The third equality holds by using that there exists a unique $A_j \in \mathcal{S}(y)$ such that $C \subseteq A_j$ by Lemma 2. Under the assumption that $|\mathcal{S}(y \cap N(C))| > |\mathcal{S}((y \ominus C) \cap N(C))|$, it must be the case that $\mu(G'[A_j \setminus C]) = 0$. Thus, we have that $A_j \setminus C = \varnothing$ which implies $A_j \subseteq C$ and so $A_j = C$.

Next consider the case when $C \in \mathcal{S}_y(\hat{y}_-)$ is negatively critical due to $|\mathcal{S}(y \cap N(C))| < |\mathcal{S}((y \ominus C) \cap N(C))|$. Again using the identity in Equation 2, the assumed inequality implies that $\mu(G'[A_j \setminus C]) \geq 2$ and so $G'[A_j \setminus C]$ must contain at least two connected components. Thus, this set can be decomposed into connected components such that

$$
A_j \setminus C = \bigcup_{k=1}^{K} B_k \subseteq \mathcal{S}((y \ominus C) \cap N(C))
$$

with $K \geq 2$. For any $v_0 \in B_1$ and $v_k \in B_2$, it is impossible to construct a path between these vertices that does not pass through $C$. Otherwise, this would imply that $B_1$ and $B_2$ are path-connected in the graph $G'[A_j \setminus C]$ and not distinct connected components.

($\Leftarrow$) Assume that Condition 1 holds, then $\exists A_j \in \mathcal{S}(y \cap N(C))$ such that $A_j = C$. The follows immediately by

$$
\begin{aligned}
|\mathcal{S}((y \ominus C) \cap N(C))| &= |\mathcal{S}(y \cap N(C))| - 1 + \mu(G'[A_j \setminus C]) \\
&= |\mathcal{S}(y \cap N(C))| - 1 + \mu(G'[C \setminus C]) \\
&= |\mathcal{S}(y \cap N(C))| - 1 + \mu(G'[\varnothing]) \\
&= |\mathcal{S}(y \cap N(C))| - 1
\end{aligned}
$$

$$
\implies |\mathcal{S}((y \ominus C) \cap N(C))| < |\mathcal{S}(y \cap N(C))|.
$$

Now assume that Condition 2 holds, then there exists distinct components $B_1, B_2 \in \mathcal{S}((y \ominus C) \cap N(C))$ with $B_1, B_2 \subset A \setminus C$ such that $v_0 \in B_1$ and $v_k \in B_2$. Since $B_1, B_2 \subset A \setminus C$ are distinct components in the graph $G'[A \setminus C]$, the final result holds by

$$
\begin{aligned}
|\mathcal{S}((y \ominus C) \cap N(C))| &= |\mathcal{S}(y \cap N(C))| - 1 + \mu(G'[A_j \setminus C]) \\
&\geq |\mathcal{S}(y \cap N(C))| - 1 + \mu(G'[B_1] \cup G'[B_2]) \\
&= |\mathcal{S}(y \cap N(C))| - 1 + \mu(G'[B_1]) + \mu(G'[B_2]) \\
&= |\mathcal{S}(y \cap N(C))| + 1
\end{aligned}
$$

$$
\implies |\mathcal{S}((y \ominus C) \cap N(C))| > |\mathcal{S}(y \cap N(C))|.
$$

$\square$

## A.2 SPECIAL CASE

**Lemma 3.** *A component $C \in \mathcal{S}_y(\hat{y}_-)$ is negatively critical with if and only if there exists an $A \in \mathcal{S}(y \cap N(C))$ with $A \supseteq C$ such that either: (1) $A = C$ or (2) $\exists v_0, v_k \in A \setminus C$ such that there does not exist a path $(v_0, \ldots, v_k) \subseteq N(C)$ with $v_i \notin C$ for $i = 1, \ldots, k-1$.*

*Proof.* The forward direction holds by applying the same argument use to prove Theorem 1. For the converse, we can again apply the same argument to prove that $|\mathcal{S}(y)| \neq |\mathcal{S}(y \ominus C)|$ which then implies that $C$ is negatively critical. $\square$

**Lemma 4.** *Given a component $C \in \mathcal{S}_y(\hat{y}_-)$ and $A \in \mathcal{S}(y)$ with $A \supseteq C$, $\exists v_0, v_k \in A$ such that there does not exist a path $(v_0, \ldots, v_k) \subseteq A \setminus C$ with $v_i \notin C$ for $i = 1, \ldots, k - 1$ if and only if $\exists B'_1, B'_2 \in \mathcal{S}(y \ominus C)$ with $B'_1, B'_2 \subset A$ such that $B'_1 \cup C \cup B'_2$.*

*Proof.* ($\Rightarrow$) It must be the case that $\hat{y}_{v_0} = y_{v_0}$ since $\{v_0, v_1\} \in E$ and $v_1 \in C$. This implies that $(y \ominus C)_{v_0} \neq 0$ and so there exists some $B'_1 \in \mathcal{S}(y \ominus C)$ with $v_0 \in B'_1$. By the same argument, $(y \ominus C)_{v_k} \neq 0$ and there exists some $B'_2 \in \mathcal{S}(y \ominus C)$ with $v_k \in B'_2$. Moreover, $v_0$ and $v_k$ must belong to distinct component, i.e. $B'_1 \neq B'_2$, since these nodes are not connected in the subgraph induced by $y \ominus C$. Lastly, $B'_1 \cup C \cup B'_2$ is connected due to the existence of the path $(v_0, \ldots, v_k)$ from $B'_1$ to $B'_2$ via $C$.

($\Leftarrow$) The converse holds immediately since $B'_1$ and $B'_2$ are disjoint by definition. $\square$

**Lemma 5.** *Given a component $C \in \mathcal{S}_y(\hat{y}_-)$ and $A \in \mathcal{S}(y)$ with $A \supseteq C$, $\exists B'_1, B'_2 \in \mathcal{S}(y \ominus C)$ with $B'_1, B'_2 \subset A$ such that $B'_1 \cup C \cup B'_2$ is connected if and only if $\exists B_1, B_2 \in \mathcal{S}(y \ominus \hat{y}_-)$ with $B_1, B_2 \subset A$ such that $B_1 \cup C \cup B_2$ is connected.*

*Proof.* ($\Rightarrow$) Given that $B'_1 \cup C \cup B'_2$ is connected, there exists a path $(v_0, \ldots, v_k)$ from $B'_1$ to $B'_2$ via $C$ since $B'_1$ and $B'_2$ are disjoint. It must be the case that $\hat{y}_{v_0} = y_{v_0}$ since $\{v_0, v_1\} \in E$ and $v_1 \in C$. This implies that $(y \ominus C)_{v_0} \neq 0$ and so $\exists B_1 \in \mathcal{S}(y \ominus \hat{y}_-)$ with $v_0 \in B_1$. The same argument can be applied to $v_k$ to prove $\exists B_2 \in \mathcal{S}(y \ominus \hat{y}_-)$ with $v_k \in B_2$. Thus, the same path that connects the sets $B'_1, B'_2$, and $A$ also connects $B_1, B_2$, and $A$.

($\Leftarrow$) The converse holds by applying the same argument. $\square$

**Corollary 1.** *A component $C \in \mathcal{S}_y(\hat{y}_-)$ is negatively critical with $|\mathcal{S}(y)| \neq |\mathcal{S}(y \ominus C)|$ if and only if there exists an $A \in \mathcal{S}(y)$ with $A \supseteq C$ such that either: (1) $A = C$ or (2) $\exists B_1, B_2 \in \mathcal{S}(y \ominus \hat{y}_-)$ with $B_1, B_2 \subset A$ such that $B_1 \cup C \cup B_2$ is connected.*

*Proof.* This result nearly follows immediately by applying Lemmas 3 - 5. $\square$

**Corollary 2.** *A component $C \in \mathcal{S}_y(\hat{y}_+)$ is positively critical with $|\mathcal{S}(\hat{y})| \neq |\mathcal{S}(\hat{y} \ominus C)|$ if and only if there exists an $A \in \mathcal{S}(\hat{y})$ with $A \supseteq C$ such that either: (1) $A = C$ or (2) $\exists B_1, B_2 \in \mathcal{S}(\hat{y} \ominus \hat{y}_+)$ with $B_1, B_2 \subset A$ such that $B_1 \cup C \cup B_2$ is connected.*

*Proof.* Let $z = \hat{y}$ and $\hat{z}_- = \hat{y}_+$, then the result follows immediately by applying Corollary 1. $\square$

### A.3 COMPUTATION

**Corollary 3.** *A component $C \in \mathcal{S}_y(\hat{y}_-)$ is negatively critical with $|\mathcal{S}(y)| \neq |\mathcal{S}(y \ominus C)|$ if and only if $\exists A \in \mathcal{S}(y)$ with $A \supseteq C$ such that either: (1) $\nexists i \in D(C)$ with $i \in A$ or (2) $\exists B_1, B_2 \in \mathcal{S}(y \ominus \hat{y}_-)$ with $B_1, B_2 \subset A$ such that $i \in B_1$ and $j \in B_2$ for some $i, j \in D(C)$.*

*Proof.* First, we prove that Condition 1 is equivalent to Condition 1 in Corollary 1. For the forward direction, Lemma 2 implies that $C \subseteq A$. Now choose any $i \in A$, then it must be the case that $i \in C$ since $A$ is connected and $\nexists j \in D(C)$ with $y_i = y_j$. The converse is trivial since $A = C$ implies that

$$D(C) = N(C) \setminus C = N(A) \setminus A$$

and so $\nexists i \in D(C)$ with $i \in A$ since the set $A$ is entirely removed from $D(c)$.

Next, we prove that Condition 2 is equivalent to Condition 2 in Corollary 1. Let $v_0 = i$ and $v_k = j$, there exists a path connecting these vertices contained in $C$ since these nodes are connected to the boundary of $C$ which is a connected set. For the converse, $B_1 \cup C \cup B_2$ being connected but $B_1 \cup B_2$ being disconnected implies that there exists a path $(v_0, \ldots, v_k)$ from $B_1$ to $B_2$ via $C$. Since this path passes through $C$, it must be the case that $\exists v_i, v_j \in D(C)$ such that $v_i \in B_1$ and $v_j \in B_2$. $\square$

**Theorem 2.** *The computational complexity of computing critical components is $\mathcal{O}(n)$ with respect to the number of voxels in the image.*

*Proof.* This algorithm involves first precomputing the false negative and false positive masks which can be computed in linear time by comparing each entry in the prediction and ground truth. Next, we must also precompute the following sets of connected components: $\mathcal{S}(y)$, $\mathcal{S}(\hat{y})$, $\mathcal{S}(y \ominus \hat{y}_-)$, and $\mathcal{S}(\hat{y} \ominus \hat{y}_+)$. Since connected components can be computed in linear time, these precomputations can also be done in linear time Cormen et al. (2009).

Next, a BFS is performed over both $\hat{y}_-$ and $\hat{y}_+$ to extract the connected components of the false negative and/or positive masks. During this BFS, we can determine whether a component satisfies Corollary 3 in lines 13-21 in Algorithm 2. A BFS is a linear time algorithm Cormen et al. (2009). Since the operations in lines 13-18 can be achieved in constant time, the complexity of this BFS is still linear. $\square$

## B    EXTENSION TO AFFINITY MODELS

An affinity model is a graph-based segmentation model where the main objective is to determine whether neighboring nodes belong to the same segment. Given a graph $G = (V, E)$ and ground truth segmentation $y$, let $\delta : E \to \{0, 1\}$ be the affinity function given by

$$\delta(\{i, j\}) = \begin{cases} 1 & \text{if } i, j \in A \\ 0 & \text{otherwise} \end{cases}$$

for some $A \in \mathcal{S}(y)$. One important advantage of affinity models is that instance segmentation can be equivalently formulated as a binary classification task. This property is especially useful when the number of segments is unknown and distinct objects may touch.

In recent years, neural networks have been successfully used to learn edge affinities (Turaga et al., 2010). This model is formulated as learning affinity channels where each channel represents the connectivity along a certain direction (e.g. vertical or horizontal). Thus, the loss function is defined as a sum over loss functions corresponding to each channel.

**Definition 4.** *Let $\mathcal{L} : \mathbb{R}^{nk} \times \mathbb{R}^{nk} \to \mathbb{R}$ be the topological loss function for an affinity-based model with $k$ channels be given by*

$$\mathcal{L}(Y, \hat{Y}) = \sum_{i=1}^{k} \mathcal{L}_0(y^{(i)}, \hat{y}^{(i)}) + \alpha\, \mathcal{L}_-(y^{(i)}, \hat{y}^{(i)}) + \beta\, \mathcal{L}_+(y^{(i)}, \hat{y}^{(i)})$$

*such that $\alpha, \beta \in \mathbb{R}_+$ and $\mathcal{L}_0$ is an arbitrary loss function (e.g. cross entropy or Dice coefficient).*

Affinity-based models involve a transformation between voxel and edge-based representations of an image. It is important to note that critical components are defined with respect to the voxel-based representation of an image. This means that training an affinity-based model with our topological loss function involves performing this transformation after generating each prediction.

Although each prediction can be directly transformed to voxels, it is more computationally efficient to first compute the false negative and positive mask of each channel and then perform the transformation. The reason being that computing connected components in linear in the number of voxels in the foreground. Since the false negative and positive masks can be computed in constant time and typically have significantly fewer foreground voxels, it's much faster to compute connected components of these masks as opposed to the prediction.

## C    EXPERIMENTS

### C.1    TRAINING PROTOCOL

For the neuron segmentation experiments, we trained a U-Net performs segmentation by learning 3-d affinities (see Appendix B for more details), then apply a watershed-based algorithm that agglomerates a 3D over-segmentation computed from the 3D affinity prediction. In

this experiment, we implemented these methods by using the following Github repositories: https://github.com/jgornet/NeuroTorch and https://github.com/funkey/waterz.

Each model was trained for a total of 1500 epochs with a learning rate of $10^{-3}$ and batch size of 8. For the topological loss functions, we used the first 300 epochs to train a baseline model, then fine-tuned the model for the last 1200 epochs.

## C.2 EVALUATION METRICS

### C.2.1 NEURON SEGMENTATION

The main objective of neuron segmentation is to reconstruct the morphology of individual neurons and uncover the connectivity between them. Following segmentation, each object is skeletonized to produce a graphical representation of the neuron. Thus, we use skeleton-based metrics, rather than voxel-based metrics, to evaluate the performance of each model since the final result is a skeleton (Januszewski et al., 2017).

Let $S_i = (V_i, E_i)$ be an undirected graph that represents the skeleton of the $i$-th object in a ground truth segmentation $y$. Let $\{S_1, \ldots, S_n\}$ be a collection of skeletons such that there exists is a bijection between skeletons and objects in the segmentation. Let $V_i$ be the vertex set and assume that a node $u \in V_i$ is defined by the 3-d coordinates $\varphi(u)$. Given a node $u \in V_i$, let $y[\varphi(u)]$ be the label of node $u$ in the ground truth segmentation. Let $\hat{y}$ be the predicted segmentation to be evaluated.

A *split* is a pair of nodes $u_1, u_2 \in V_i$ such that $\hat{y}[\varphi(u_1)] \neq \hat{y}[\varphi(u_2)]$ and either (i) $\{u_1, u_2\} \in E_i$ or (ii) there exists a path $(u_1, w_1, \ldots, w_t, u_2)$ such that $w_r = 0$ for all $r = 1, \ldots, t$. The metric **# Splits** is the number of splits in the collection of ground truth skeletons $\{S_1, \ldots, S_n\}$,

$$\textbf{\# Splits} = \sum_{i=1}^{n} \left| \left\{ \{u_1, u_2\} \in V_i^2 : u_1, u_2 \text{ is a split} \right\} \right|$$

An edge $\{u, v\} \in E_i$ is said to be *omit* if (i) $\hat{y}[\varphi(u)] = 0$ or $\hat{y}[\varphi(v)] = 0$ and (2) there exists a path $(w_s, \ldots, w_1, u, v, w_1', \ldots, w_t')$ such that $w_s, w_t'$ is a split where $w_r = 0$ for all $r = 1, \ldots, s - 1$ and $w_r' = 0$ for all $r = 1, \ldots, t - 1$. The metric **% Omit** is the percentage of edges that are omit from the ground truth,

$$\textbf{\% Omit} = 100 \cdot \frac{\sum_{i=1}^{n} \left| \{e \in E_i : e \text{ is omit}\} \right|}{\sum_{i=1}^{n} |E_i|}$$

Note that one challenge of detecting omit edges is that this criteria is sensitive to minor misalignments between the ground truth skeleton and predicted segmentation mask. The purpose of the second criteria is to prevent misalignments from being detected as omits.

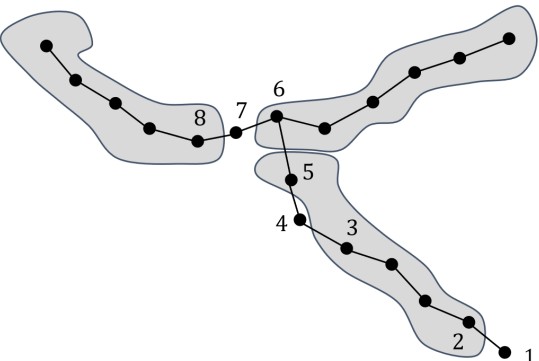

Figure 6: Here we see a single ground truth skeleton and predicted segmentation mask that consists of three connected components. Edge $\{1, 2\}$ is omit but not considered a split. Edges $\{3, 4\}$ and $\{4, 5\}$ are not considered to be omit, these edges are only slightly misaligned with the predicted segmentation. Edge $\{5, 6\}$ is a split. Edges $\{6, 7\}$ and $\{7, 8\}$ are both omits. In addition, there is a split between nodes 6 and 8.

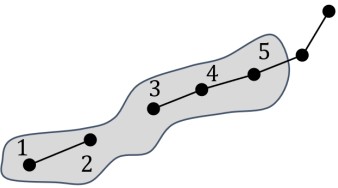

Figure 7: In this example, we see that edge $\{1, 2\}$, $\{3, 4\}$, and $\{4, 5\}$ are merged.

Let $\hat{S}_i = (\hat{V}_i, \hat{E}_i)$ be an undirected graph that represents the skeleton of the $i$-th object in a predicted segmentation $\hat{y}$. Let $\{\hat{S}_1, \ldots, \hat{S}_m\}$ be a collection of skeletons such that there exists is a bijection between skeletons and objects in the segmentation.

A *merge* is a pair of nodes $u_1, u_2 \in \hat{V}_i$ such that $y[\varphi(u_1)] \neq y[\varphi(u_2)]$ and either (i) $\{u_1, u_2\} \in \hat{E}_i$ or (ii) there exists a path $(u_1, w_1, \ldots, w_t, u_2)$ such that $w_r = 0$ for all $r = 1, \ldots, t$. The metric **# Merges** is the number of merges in the collection of predicted skeletons $\{\hat{S}_1, \ldots, \hat{S}_n\}$,

$$\textbf{\# Merges} = \sum_{i=1}^{m} \left| \left\{ \{u_1, u_2\} \in \hat{V}_i^2 : u_1, u_2 \text{ is a merge} \right\} \right|$$

Note that the definition of a merge is nearly identical to the definition of a split. The difference is that splits are detected by comparing the ground truth skeletons and predicted segmentation. In contrast, merges are detected by comparing the predicted skeletons and ground truth segmentation.

An edge $\{u_j, v_j\} \in E_i$ is *merged* if these exists an edge $\{u_k, v_k\} \in E_i$ such that $\hat{y}[\varphi(u_i)] = \hat{y}[\varphi(u_k)]$ and $\hat{y}[\varphi(v_i)] = \hat{y}[\varphi(v_k)]$ but $y[\varphi(u_i)] \neq y[\varphi(u_k)]$ and $y[\varphi(v_i)] = y[\varphi(v_k)]$. The metric **% Merged** is the fraction of edges that are merged,

$$\textbf{\% Merged} = 100 \cdot \frac{\sum_{i=1}^{n} \left| \{e \in E_i : e \text{ is merged}\} \right|}{\sum_{i=1}^{n} |E_i|}$$

The metric **Edge Accuracy** is the fraction of correctly reconstructed edges in the ground truth skeletons,

$$\textbf{Edge Accuracy} = 100 - (\textbf{\% Omit} + \textbf{\% Merged})$$

### C.2.2 VESSEL SEGMENTATION

We use the following metrics to evaluate our proposed loss function in the vessel segmentation task:

**Accuracy**: Fraction of correctly labeled voxels.

**Dice**: Metric that combines precision and recall into a single value to provide a balanced measure of a model's performance.

**Adapted Rand Index (ARI)**: Metric used to measure the similarity between two clusterings by comparing the agreement and disagreement in pairwise assignments. In this version of the Rand, index zero is excluded as a component.

**Variation of Information (VOI)**: Variation of Information (VOI): distance measure between two clusterings by measuring the amount of information lost or gained when transitioning from one clustering to another.

### C.2.3 ADDITIONAL EXPERIMENTAL RESULTS

Figure 5 from Section 4 shows qualitative results on image patches from the DRIVE dataset. In Figure 8, we show the predicted segmentations on the full images.

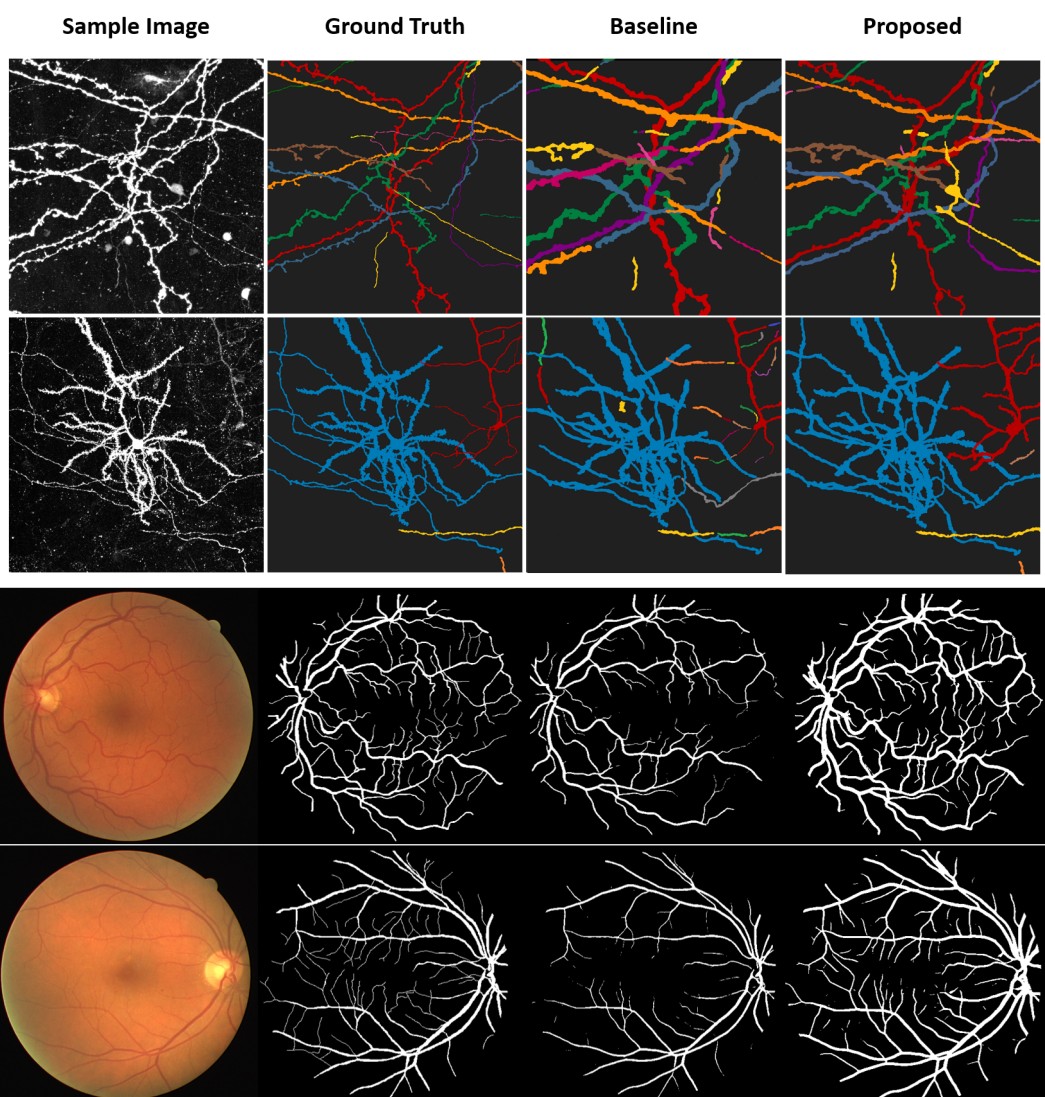

Figure 8: Qualitative results of the proposed method on the EXASPIM and DRIVE dataset. Baseline is a U-Net trained with cross entropy and Proposed is a U-Net trained with our supervoxel loss.

Table 3: Quantitative results on individual blocks from the EXASPIM dataset

| Method | Block Dimensions | # Splits | % Omitted | # Merges | % Merged | % Accuracy |
|---|---|---|---|---|---|---|
| Cross Entropy | (512, 1024, 1024) | 23 | 2.61 | 0 | 0 | 97.39 |
| | (1024, 1024, 1024) | 78 | 20.71 | 8 | 27.19 | 52.10 |
| | (1024, 1024, 1024) | 27 | 4.20 | 2 | 13.16 | 82.64 |
| | (1024, 1024, 1024) | 30 | 13.23 | 6 | 12.36 | 74.41 |
| Dice | (512, 1024, 1024) | 14 | 5.50 | 0 | 0 | 94.50 |
| | (1024, 1024, 1024) | 92 | 38.31 | 7 | 32.92 | 28.87 |
| | (1024, 1024, 1024) | 13 | 7.96 | 0 | 0 | 92.04 |
| | (1024, 1024, 1024) | 34 | 15.66 | 9 | 23.51 | 60.83 |
| clDice | (512, 1024, 1024) | 2 | 0.86 | 0 | 0 | 99.24 |
| | (1024, 1024, 1024) | 126 | 16.69 | 4 | 8.13 | 75.18 |
| | (1024, 1024, 1024) | 11 | 5.11 | 1 | 7.17 | 87.72 |
| | (1024, 1024, 1024) | 30 | 3.33 | 5 | 9.21 | 87.46 |
| Simple Voxel | (512, 1024, 1024) | 22 | 3.54 | 0 | 0 | 96.46 |
| | (1024, 1024, 1024) | 107 | 16.62 | 6 | 4.83 | 78.55 |
| | (1024, 1024, 1024) | 9 | 5.26 | 1 | 17.01 | 94.57 |
| | (1024, 1024, 1024) | 22 | 10.29 | 13 | 2.42 | 87.29 |
| MALIS | (512, 1024, 1024) | 4 | 0.50 | 0 | 0 | 99.50 |
| | (1024, 1024, 1024) | 82 | 14.63 | 6 | 6.32 | 79.05 |
| | (1024, 1024, 1024) | 13 | 5.31 | 1 | 5.58 | 89.11 |
| | (1024, 1024, 1024) | 22 | 9.55 | 5 | 8.54 | 81.91 |
| **Ours** | (512, 1024, 1024) | 1 | 0.90 | 0 | 0 | 99.10 |
| | (1024, 1024, 1024) | 50 | 7.83 | 8 | 9.21 | 82.96 |
| | (1024, 1024, 1024) | 6 | 2.86 | 0 | 0 | 97.14 |
| | (1024, 1024, 1024) | 10 | 6.41 | 6 | 7.67 | 85.92 |