# OpenReview forum: "Instance Segmentation with Supervoxel Based Topological Loss Function"
_ICLR.cc/2024/Conference — Submitted to ICLR 2024_

### Official Review · Reviewer_cPYQ · 2023-11-01

**Soundness:** 3 good
**Presentation:** 4 excellent
**Contribution:** 2 fair
**Rating:** 5
**Confidence:** 4

**Summary:**

The authors work on binary as well as instance segmentation of curvilinear structures. This segmentation task is prone to errors such as split and merge mistakes. The authors resolve such errors by extending the topological concept of simple points to superpixels (or supervoxels). While most topological-based methods are computationally expensive, the authors propose an algorithm to reduce the complexity of their method. The authors validate their method on 2 datasets and compare against several topology-aware baselines.

**Strengths:**

1) The authors extend the concept of simple points to supervoxels. In the false negative and false positive maps, they check if keeping or removing the connected component (CC) changes the topology or not. If it changes the topology, then they deem it critical and apply high weight to it in the loss function while training. The critical CCs correspond to the split/merge errors that one would like to resolve.
2) The authors develop an $O(n)$ solution where $n$ is the number of pixels/voxels using standard graph algorithms like BFS. This is much cheaper compared to existing topology-aware methods whose algorithms are atleast $O(n \log n)$ or $O(n^2)$.
3) The authors provide adequate proofs of their runtime in the supplementary.

**Weaknesses:**

1) In principle, the novelty of the contribution seems limited as an existing concept of simple points has been extended to superpixels (collection of pixels) instead.
2) The authors should consider comparing against clDice [1] as a baseline since clDice has shown better performances among the topology-aware methods.
3) The authors do not provide any ablation study. Considering they have hyperparameters $\alpha$ and $\beta$ in their loss function, the authors would benefit from providing an ablation study of these loss weights and provide a discussion on how each hyperparameter affects the results.

**References**

[1] Shit, Suprosanna, et al. "clDice-a novel topology-preserving loss function for tubular structure segmentation." Proceedings of the IEEE/CVF Conference on Computer Vision and Pattern Recognition. 2021.

**Questions:**

1) Please also see the weakness above.
2) As the authors claim that they are proposing a topology-aware neural network, they should also evaluate the result on topology-aware metrics like clDice [1], Betti Matching [2], and Betti Number [3].
3) Please mention if the numbers in bold are just numerically better, or, if t-test [4] has been conducted to check if the performance improvement is statistically significant or not.

**References**

[1] Shit, Suprosanna, et al. "clDice-a novel topology-preserving loss function for tubular structure segmentation." Proceedings of the IEEE/CVF Conference on Computer Vision and Pattern Recognition. 2021.

[2] Stucki, Nico, et al. "Topologically faithful image segmentation via induced matching of persistence barcodes." International Conference on Machine Learning. PMLR, 2023

[3] Hu, Xiaoling, et al. "Topology-preserving deep image segmentation." Advances in neural information processing systems 32 (2019).

[4] Student, 1908. The probable error of a mean. Biometrika, pp.1–25.

---

> ### Author Response · Authors · 2023-11-18
> **Response to Weaknesses**
>
> W1: In principle, the novelty of the contribution seems limited as an existing concept of simple points has been extended to superpixels (collection of pixels) instead.
>
> A1: While the proposed extension is indeed simple to state (i.e., extension of the concept of simplicity from voxels to supervoxels), it took us a while to conceptualize this. We would like to note that the concepts on which our work is built (e.g., simple pixel, supervoxels) has been around for decades and there is an ever-growing need for topology-preserving methods. Therefore, we believe the proposed extension appears simple (and potentially limited) only after the fact. Finally, this extension would not be valuable unless it is supported by an accompanying mathematical and algorithmic framework. We are not aware of shorter, simpler ways of developing a rigorous theory without going through the multiple formal statements in our manuscript. Please also refer to our general response on this.
>
> W2: The authors should consider comparing against clDice [1] as a baseline since clDice has shown better performances among the topology-aware methods.
>
> A2: We are actively working on training a model with the clDice. However, the original code base has some issues, as documented on the official github repository, which are delaying us from obtaining results. If we can resolve these, we aim to report the results of training this model to perform instance segmentation on the EXASPIM23 dataset.
>
> W3: The authors do not provide any ablation study. Considering they have hyperparameters and in their loss function, the authors would benefit from providing an ablation study of these loss weights and provide a discussion on how each hyperparameter affects the results.
>
> A3: Thanks for this comment. Indeed, this important point was inadvertently left out in the initial submission. We have performed hyperparameter optimization beyond what's reported in the original submission to elucidate the sensitivity of performance on hyperparameter values. We found that the performance changes less than 10\% across changes of a few orders of magnitude around the optimal values of the hyperparameters $\alpha$ and $\beta$. Thus, these experiments suggest that careful hyperparameter tuning is not necessary in practice. We will add both the figure describing these findings and a surrounding discussion to the revision.

---

> ### Author Response · Authors · 2023-11-18
> **Responses to Questions**
>
> Q1: As the authors claim that they are proposing a topology-aware neural network, they should also evaluate the result on topology-aware metrics like clDice [1], Betti Matching [2], and Betti Number [3].
>
> A1: We have now added an additional metric, the Betti error, to better quantify the results reported for the DRIVE dataset, as suggested. Briefly, the results support our previous findings where the proposed method is a close second-best for this metric and the top performer in other previously reported metrics, despite focusing on a narrower set of topological changes and the ensuing computational simplicity.
>
> Q2: Please mention if the numbers in bold are just numerically better, or, if t-test [4] has been conducted to check if the performance improvement is statistically significant or not.
>
> A2: The bold numbers highlight the best value across all methods. The paper has been updated to indicate this.

---

> ### Comment · Reviewer_cPYQ · 2023-11-21
>
> Could the authors provide the quantitative results of the answers mentioned in the above response (weakness A3 and questions A1)? Either as comments here or in the updated PDF.
>
> Considering the proposed method is a close-second on the Betti error metric, which method was the best? Considering this is a topology-based approach, we should see significant improvement in topology-aware metrics.

---

> > ### Author Response · Authors · 2023-11-21
> > **See updated manuscript**
> >
> > Q1: Could the authors provide the quantitative results of the answers mentioned in the above response (weakness A3 and questions A1)? Either as comments here or in the updated PDF.
> >
> > A1a: We have uploaded an updated version of our manuscript that includes a paragraph on hyperparameter optimization in the experiments section.
> >
> > A1b: We have updated Table 2 to include the Betti error of each model on the retina vessel segmentation task. The results of this experiment are that DMT [1] has the best performance with an average Betti error of $0.873\pm0.402$. Our model is the next best performance with an average Betti error of $0.944\pm0.269$, which outperforms the baseline and three models that also utilize a topological loss function (i.e. [2], [3], [4]).
> >
> >
> > [1] Xiaoling Hu, Yusu Wang, Fuxin Li, Dimitris Samaras, and Chao Chen. Topology-aware segmenta-
> > tion using discrete morse theory. In International Conference on Learned Representations (ICLR),
> > 2021.
> >
> > [2] Ahmed Fakhry, Hanchuan Peng, and Shuiwang Ji. Deep models for brain em image segmentation:
> > novel insights and improved performance. Bioinformatics, 32(15):2352–2358, 2016.
> >
> > [3] Xiaoling Hu, Fuxin Li, Dimitris Samaras, and Chao Chen. Topology-preserving deep image seg-
> > mentation. In Advances in Neural Information Processing Systems (NeurIPS), volume 32, pp.
> > 5657–5668, 2019.
> >
> > [4] Agata Mosinska, Pablo Marquez-Neila, Mateusz Kozinski, and Pascal Fua. Beyond the pixel-wise
> > loss for topology-aware delineation. In 2018 IEEE/CVF Conference on Computer Vision and
> > Pattern Recognition (CVPR), pp. 3136–3145, 2018.

---

### Official Review · Reviewer_ZGQy · 2023-11-02

**Soundness:** 2 fair
**Presentation:** 2 fair
**Contribution:** 2 fair
**Rating:** 5
**Confidence:** 5

**Summary:**

This paper proposed a "supervoxel based topological loss function" to solve the connectivity-related problems in segmentation tasks. The paper considered the key components of false positives and false negatives as key factors affecting the topology and uses loss functions to optimize them. Through algorithm design, the time complexity is reduced. The theoretical proof is rich, and the effectiveness of the model is verified on EXASPIM2 and DRIVE. The visualization effect shows that the loss function proposed in this paper can effectively improve the visualization effect.

**Strengths:**

1. The paper proposed novel ideas and methods that are simple and effective after being optimized by the proposed algorithm in this article.
2. The paper verified the effectiveness of the method in both 3D and 2D dimensions.

**Weaknesses:**

1. Lack of visual comparison with baseline in Figure 5.
2. Performance metrics should be described using formulas.
3. Lack of comparison with more loss functions that can supervise topology changes.

**Questions:**

Critical components should not include areas that do not affect the topology structure. In the visualization effect of Figure 4, why is the structure more slender compared to the baseline, and can the proposed loss function optimize the segmentation edge?

What are the values of α and β in the experiment?

---

> ### Author Response · Authors · 2023-11-18
> **Responses to Weaknesses**
>
> W1: Lack of visual comparison with baseline in Figure 5.
>
> A1: We have now added the visual comparison to Figure 5, as suggested by this referee.
>
> W2: Performance metrics should be described using formulas.
>
> A2: We have now added equations describing the performance metrics to the Supplementary material and referred to them in the main text when the metrics are introduced.
>
> W3: Lack of comparison with more loss functions that can supervise topology changes.
>
> A3: We have now added an additional metric, the Betti error, to better quantify the results reported for the DRIVE dataset, as suggested. Briefly, the results support our previous findings where the proposed method is a close second-best for this metric and the top performer in other previously reported metrics, despite focusing on a narrower set of topological changes and the ensuing computational simplicity.

---

> ### Author Response · Authors · 2023-11-18
> **Responses to Questions**
>
> Q1: Critical components should not include areas that do not affect the topology structure. In the visualization effect of Figure 4, why is the structure more slender compared to the baseline, and can the proposed loss function optimize the segmentation edge?
>
> A1:We attribute this qualitative difference to the fact that a less slender structure is more likely to result in merge mistakes. The depiction of positively critical components in Figure 3 is a didactic cartoon and might not accurately portray a "typical" critical component. In actuality, positively critical components encompass incorrect voxel predictions that act as a bridge between two distinct components. (Plus, any voxels in the false positive mask connected to this bridge which may extend into a large region connected to the boundary of the objects.) It's plausible that the neural network learns to avoid less slender predictions due to these considerations.
>
> We don't make any claims that this loss function leads to more slender predictions. For neuron segmentation, this is a less important qualitative feature and not easy to quantify.

---

### Official Review · Reviewer_Jbm6 · 2023-11-08

**Soundness:** 2 fair
**Presentation:** 2 fair
**Contribution:** 3 good
**Rating:** 6
**Confidence:** 4

**Summary:**

Traditional segmentation methods focus on total voxel accuracy. While some critical voxel errors might change the topology, most would not. This paper proposes a method to find those critical voxels and add additional penalty terms to the loss function whenever these voxels are inaccurately predicted.

The contributions of this paper are as follows:
1. An algorithm is proposed to detect split and merge errors which disrupt the number of components relative to the ground truth.
2. The computational complexity of this algorithm scales linearly with the number of voxels under the assumption of a tree graph.
3. This approach can be easily integrated into available segmentation frameworks as it is built on top of common voxel-based loss functions.
4. The experiments conducted show the effectiveness of the proposal relative to competing methods on two datasets.

**Strengths:**

In addition to the contributions listed above I would highlight the following:

1. The authors provide a good mathematical notation to communicate their methodology.
2. The didactic images facilitate a better understanding of the approach.
3. Details of training are explicitly mentioned such as the continuation scheme.

**Weaknesses:**

1. The paper could improve in terms of clarity in several cases. Most importantly, Algorithm 2--a critical part of the paper--is very vague and the only explanation provided about it under Corollary 3 does little to make it clearer.
2. More emphasis needs to be placed that the O(n) gurantee is only valid if critical components affect both local and global topology, i.e. having a tree graph.
3. The method is only sensitive to the number of components, while topology is much broader e.g. bifurcations, loops. This limitation needs to be communicated.
4. The approach comes with hyper-parameters which might be time consuming to set.

**Questions:**

1. The loss function is introduced in Definition 1, and then expanded again in Section 3.3. A better sense of direction would have been conveyed if the two were mixed and mentioned early on in Section 3 and stated that the rest of the Section focuses on finding N(y^) and P(y^) in Definition 4. Removing G(.) and H(.) and having double sums might improve readablility.

2. The definition originally provided for S(.) does not have a subscript and is clear, but the explanation provided for the case with a subscript is hard to grasp.

3. In Method Section before Definition 1, it is stated that y_i \in {0,1,...,m} and y^_i \in {0,1,...,l}. Shouldn't they both sets be the same (no need for l)? Also m, l(?), and n are given without definitions.

4. Positively critical components rely on computing S(y^⊖y^+), while Algorithm 2 only requries knowing S(y⊖y^-). Why?

5. Condition 1 and Condition 2 are not really defined as such.

6. Potential typos: Section 3.1.2 line 2: false negative mask -> false positive mask; Two lines above Corollary 1: lemma seems extra.

---

> ### Author Response · Authors · 2023-11-18
> **Response to Weaknesses**
>
> W1: The paper could improve in terms of clarity in several cases. Most importantly, Algorithm 2--a critical part of the paper--is very vague and the only explanation provided about it under Corollary 3 does little to make it clearer.
>
> A1: Thank you for this very helpful and explicit feedback. In retrospect, we agree that the initial presentation of this section may be difficult to parse. We have completely rewritten this section to improve clarity and focus the discussion on the high level ideas. The pseudo code has as been significantly revised so that it is more intuitive and better aligned to the descriptions in the text.
>
> W2+W3: The method is only sensitive to the number of components, while topology is much broader e.g. bifurcations, loops. This limitation needs to be communicated. More emphasis needs to be placed that the O(n) guarantee is only valid if critical components affect both local and global topology, i.e. having a tree graph.
>
> A2+A3: While the main definitions and initial formal statements (e.g., Thm 1) capture topological changes in structures broader than tree-like objects, as this referee pointed out, the statements and algorithms surrounding the fast implementation are restricted to tree-structured objects. Indeed, a similar algorithm based on the main definitions and deductions can be implemented in a straightforward way, except that this algorithm will be super-linear in complexity. To address the referee's concern, we have now added explanation to the main text (see discussion following Theorem 3) emphasizing that the proposed fast algorithm will not be sensitive to broader topological changes in more general graph structures, such as creation of a cavity or a loop.
>
> W4: The approach comes with hyper-parameters which might be time consuming to set.
>
> A4: Thanks for this comment. Indeed, this important point was inadvertently left out in the initial submission. We have performed hyperparameter optimization beyond what's reported in the original submission to elucidate the sensitivity of performance on hyperparameter values. We found that the performance changes less than 10\% across changes of a few orders of magnitude around the optimal values of the hyperparameters $\alpha$ and $\beta$. Thus, these experiments suggest that careful hyperparameter tuning is not necessary in practice. We will add both the figure describing these findings and a surrounding discussion to the revision.

---

> ### Author Response · Authors · 2023-11-18
> **Response to Questions**
>
> Q1: The loss function is introduced in Definition 1, and then expanded again in Section 3.3. A better sense of direction would have been conveyed if the two were mixed and mentioned early on in Section 3 and stated that the rest of the Section focuses on finding N(y) and P(y) in Definition 4. Removing G(.) and H(.) and having double sums might improve readablility.
>
> A1: This is a great suggestion and we have updated the paper accordingly. It not only improved readability, but also shortened Section 3.3 so that we can present more visual results in Section 4.
>
>
> Q2: The definition originally provided for S(.) does not have a subscript and is clear, but the explanation provided for the case with a subscript is hard to grasp.
>
> A2: This extra notation is necessary to ensure that each connected component $C\in\mathcal S(\hat y_-)$ intersects with exactly one connected component in the set $\mathcal S(y)$. To address the referee's concern, we have now added an explanatory footnote after the definition with the subscript, relating to the definition without the subscript.
>
>
> Q3: In the Method Section before Definition 1, it is stated that $y_i \in\{0,1,...,m\}$ and $\hat y_i \in\{0,1,...,l\}$. Shouldn't they both sets be the same (no need for l)? Also m, l(?), and n are given without definitions.
>
> A3: Thank you for pointing this out, there is a typo in the manuscript. Our original submission contains the lines ``A ground truth segmentation $y = (y_1, . . . , y_n)$ is a labeling of the vertices such that $y_i\in\{0, 1,..., m\}$ denotes the label of node $i \in V$. Each segment has a label in 1,..., k and the background is marked with 0.'' Instead the last sentence should be "...Each segment has a label in 1,..., *m and the background is marked with 0", which defines m to be the number of objects in the image.
>
> The value l is the number of objects in the predicted segmentation. We assume that the true number of objects m is unknown at the time of inference. It is possible and usually the case that $l \neq m$, especially in neuron segmentation. We added a footnote explaining why m is not necessarily equal to l.
>
> We also updated the first sentence in the method section to include the definition of n, it know reads as ``Let $G=(V,E)$ be an undirected graph with the vertex set $V=\{1,\ldots, n\}$. We assume that $G$ is a graphical representation of an image where the vertices represent voxels and edges are defined with respect to a $k$-connectivity''.
>
>
> Q4: Positively critical components rely on computing $S(\hat y\ominus \hat y_+)$, while Algorithm 2 only requires knowing $S(y\ominus y_-)$. Why?
>
> A4: Thanks for pointing it out. This is actually a typo in Algorithm 2, it should read as $S(y\ominus \hat y_x)$ instead of $S(y\ominus \hat y_-)$, where $\hat y_x$ is a placeholder for $\hat y_-$ and $ \hat y_+$. We have now updated the pseudocode of the algorithm.
>
>
> Q5: Condition 1 and Condition 2 are not really defined as such.
>
> A5: We have now updated the text following Corollary 3 to clarify this point.
>
>
> Q6: Potential typos: Section 3.1.2 line 2: false negative mask -> false positive mask; Two lines above Corollary 1: lemma seems extra.
>
> A6: Thanks for pointing the typo of Section 3.1.2, we've now fixed this mistake. For the two lines above Corollary, we deleted the first line above the corollary and agree that it's a bit repetitive. We did leave the second line above this corollary because this is where we define y ominus hat y-.

---

> > ### Comment · Reviewer_Jbm6 · 2023-11-22
> >
> > Thank you for taking the comments to improve the quality of your work.
> >
> > I believe the paper now reads better with having a top-down approach. Several notations have been clarified or corrected which help greatly in understanding the details.
> >
> > While the revisions in Algorithm 2 has made it more precise, it seems like the explanations explicitly about it from the text have been removed (correct me if I am wrong). I think one could easily get lost in all the if statements and a high-level explanation in text is beneficial.
> >
> > I like the fact that the computational complexity has been emphasized in the form of a theorem. However, the tree-shape assumption needs to be stated as part of the theorem for clarity---and not in the following paragraph. I believe one cannot really do anything without making assumptions, so there is no shame in making and stating them.
> >
> > In summary, I find the paper to have improved to a level that is ready for publication.

---

> ### Author Response · Authors · 2023-11-22
> **Computation Section Update**
>
> We thank this reviewer for improving their score. We appreciate your additional feedback on Section 3.2. A revised version has been uploaded, incorporating your latest comments. We significantly revised this section to provide a higher level description of the mechanics of the algorithm. In the process, some text was indeed removed.
>
> We have further updated the paper to improve both the readability and preciseness of our algorithm description. In the most up-to-date version of the paper, we have made slight adjustments to the organization of the algorithm's description. We have also improved the clarity by explicitly indicating the lines in the algorithm to which the text description corresponds.
>
> Thank you for pointing out that the statement of Theorem 3 does not refer to a "tree-structured" condition. We have updated this theorem so that it now includes the same condition stated in Corollaries 1-3. In addition, we also refer to this condition in the discussion following this Theorem.

---

### Author Response · Authors · 2023-11-18
**General Response**

We thank the reviewers for their constructive comments that have significantly improved the quality of this work. As unanimously suggested by the reviewers, we added new experimental results on hyperparameter tuning and ablation, which further demonstrate the robustness and utility of our method. In addition, we also improved the clarity of presentation by following the reviewers' advice on emphasizing various aspects and limitations, flow of mathematical definitions, and expanding the explanation of the algorithms.

We would also like to address a potential confusion on methodological novelty. Simplicity in the statement of the core idea (i.e., extension of the concept of simplicity from voxels to supervoxels) should not imply limitedness of the conceptual advance. On the contrary, we believe post-hoc simplicity is a strength of our contribution: The notion of a simple voxel has been around for more than 30 years. Similarly, there is decades of work on connected components. Yet, despite an ever-growing need for topology-aware processing, our work is the first to propose such an extension. This may be, in part, due to difficulty in forming the right mathematical framework and supporting practicable algorithms. As the manuscript shows, it took us multiple formal statements and careful consideration of computational complexity to arrive at the proposed method.

---

### Author Response · Authors · 2023-11-21
**** Updated Manuscript ****

We would like to thank all reviewers for providing insightful and constructive feedback that we used to improve the quality of this paper. We have made a significant number of revisions and have posted an updated version of our paper. There are a number of changes that we would like to emphasize:

1. Hyperparameter Optimization

A new paragraph has been added to Section 4 on choosing $\alpha$ and $\beta$ for the neuron segmentation task. We included Figure 4 which shows the landscape of objective function values for different $\alpha$ and $\beta$ during the optimization process. Briefly, the figure demonstrates that careful tuning of the hyperparameters is not needed and performance does not change significantly within large windows of the hyperparameter set. In addition, we also provide some insights into manually tuning these parameters in the absence of using an optimization scheme.


2. Visual Comparison between Proposed Method and Baseline

In response to feedback from Reviewer ZGQy, we included additional visual comparisons between our proposed method and the baseline. Figure 5 shows four instances of image patches along with the ground truth and predicted segmentations obtained by the baseline and proposed model. These qualitative results emphasize our method's superior performance in segmentating the topology of fine structures in challenging regions within the images. In addition, we include the segmentation results on the whole image from which each patch was sampled from in Figure 8 in Appendix C.2.3.


3. Improved Computation Section

As suggested by Reviewer Jbm6, we have extensively revised the computation section. This section now emphasizes providing a high-level overview of the algorithm and articulating the underlying intuition.


4. Comparison with clDice on Neuron Segmentation

Following feedback from Reviewer cPYQ, we included a comparison between our proposed method and clDice, as shown in Tables 1 and 3. For this experiment, we trained a U-Net using clDice and configured $\alpha$ to be 0.25. The parameter $\alpha$ ranges from 0 to 0.5, and in this case, we opted for the midpoint.


5. Rigorous Definition of Skeleton-Based Metrics

In response to the feedback provided by Reviewer ZGQy, we have made substantial revisions to the evaluation metrics section found in the appendix. In contrast to our initial submission, where each skeleton-based metric was briefly summarized in a single sentence, our revised submission now includes rigorous definitions for each metric. Additionally, we created Figure 7 which visually depicts the different types of errors that are penalized by this class of metrics.

---

### Meta-Review · Area_Chair_CTd2 · 2023-12-06

**Metareview:**

This paper utilizes the notion of *simple points* from digital topology to design an instance segmentation algorithm that addresses false splits and merges of segmented curvilinear structures. This is a challenging task, also in the light of recent advances in image segmentation, as SAM has great problems segmenting curvilinear structures.

The paper's main weaknesses relate to performance and validation. In part, the baselines are still limited and do not, for instance, include recent work focusing specifically on false splits and merges in biomedical image segmentation (Lin et al, IPMI'23). In part, the performance of the model is not convincing, and the topological validation metrics are only used for the EXASPIM dataset. But measures such as Betti error or splits/merges are highly relevant also for DRIVE.

In light of these concerns, the paper is not yet ready for publication at ICLR, and I cannot recommend its acceptance. I believe, however, that the studied problem is an important one, and I encourage the authors to work on stronger and more convincing empirical evidence and resubmit to another great venue.

**Justification For Why Not Higher Score:**

There are too many concerns regarding experimental validation, and upon checking the paper I agree with the reveiwers.

**Justification For Why Not Lower Score:**

N/A

---

### Decision · Program_Chairs · 2024-01-16

Reject